# Confederated Machine Learning on Horizontally and Vertically Separated Medical Data for Large-Scale Health System Intelligence

## Abstract

A patient's health information is generally fragmented across silos. Though it is technically feasible to unite data for analysis in a manner that underpins a rapid learning healthcare system, privacy concerns and regulatory barriers limit data centralization. Machine learning can be conducted in a federated manner on patient datasets with the same set of variables, but separated across sites of care. But federated learning cannot handle the situation where different data types for a given patient are separated vertically across different organizations. We call methods that enable machine learning model training on data separated by two or more degrees "confederated machine learning." We built and evaluated a confederated machine learning model to stratify the risk of accidental falls among the elderly.

## 1 Introduction

Significance. Access to a large amount of high quality data is possibly the most important factor for success in advancing medicine with machine learning and data science. However, valuable healthcare data are usually distributed across isolated silos, and there are complex operational and regulatory concerns. Data on patient populations are often horizontally separated,each other across different practices and health systems. In addition, individual patient data are often vertically separated, by data type, across her sites of care, service, and testing. We train a confederated learning model in a manner to stratify elderly patients by their risk of a fall in the next two years, using diagnoses, medication claims data and clinical lab test records of patients. Traditionally, federated machine learning refers to distributed learning on horizontally separated data (Yue Zhao, Meng Li, Liangzhen Lai, Naveen Suda, Damon Civin, Vikas Chandra 2018; Cano, Ignacio, Markus Weimer, Dhruv Mahajan, Carlo Curino, and Giovanni Matteo Fumarola 2016; H. Brendan McMahan, Eider Moore, Daniel Ramage, Seth Hampson, Blaise Agüera y Arcas 2016; Anon n.d.). Algorithms are sent to different data silos (sometimes called data nodes) for training. Models obtained are aggregated for inference. Federated learning can reduce data duplication and costs associated with data transfer, while increasing security and shoring up institutional autonomy. (Geyer, R. C., Klein, T., Nabi, M. 2017; al. 2016),(Yue Zhao, Meng Li, Liangzhen Lai, Naveen Suda, Damon Civin, Vikas Chandra 2018; al. 2015)(Geyer, R. C., Klein, T., Nabi, M. 2017; al. 2016).

Notably, a patient's vertically separated data may span data types–for example, diagnostic, pharmacy, laboratory, and social services. Machine learning on vertically separated data has used a split neuron network (Praneeth et al. 2018) and homomorphic encryption (Praneeth et al. 2018; Stephen et al. 2017). However, these new methods require either information communication at each computational cycle or state-of-art computational resource organization, which are usually impractical in many healthcare systems where support for data analysis is not the first priority, high speed synchronized computation resources are often not available, and data availability is inconsistent.

To accelerate a scalable and collaborative rapid learning health system (Friedman et al. 2010; Mandl et al. 2014), we propose a confederated machine learning method that trains machine learning models on data both horizontally and vertically separated by jointly learning a high level representation from data distributed across silios(Qi et al. 2017; Zhang Xiao 2015; Zhai et al. 2014). This method does not require frequent information exchange at each training epoch nor state-of-the-art

distributed computing infrastructures. As such, it should be readily implementable, using existing health information infrastructure.

We demonstrate this approach by developing a model of accidental falls among people at least 65 years, a problem which causes approximately 50.7 deaths per 100,000 in the US annually (Anon 2019). Women and men from 65 to 74 years old had a 12-month fall incident rate of 42.6 and 41.3 per 100, respectively; once over 74 years old, the incident rate climbed to 50.6 and 62.0 per 100 respectively, according to the 2008 National Health Interview Survey in 2008 (Verma et al. 2016). Nationally, the direct medical costs attributable to falls is 34 billion dollars (Kramarow et al. 2015; Verma et al. 2016; Heinrich et al. 2010; Haasum Johnell 2017; Yang et al. 2016; Dollard et al. 2012; Overstall 1985; Lord et al. 2007).

There are highly effective approaches to mitigating the risk of falls that could be selectively applied to individuals identified as being at high risk. These include medication adjustments, exercises, and home interventions (McMurdo et al. 2000; Kosse et al. 2013). Multifactorial clinical assessment and management can reduce falls by more than 20

We train a confederated learning model to stratify elderly patients by their risk of a fall in the next two years, using horizontally and vertically separated diagnosis data, medication claims data and clinical lab test records of patients. The goal is to compare confederated learning with both centralized learning and traditional federated learning, and specifically test whether a confederated learning approach can simultaneously address horizontal and vertical separation.

## 2 METHODS

### 2.1 DATA SOURCE AND COHORT

The study uses claims data from a major U.S. health plan. Elements include the insurance plan type and coverage periods, age, sex, medications, and diagnoses associated with billed medical services, from July 1 2014 to June 31 2017. The dataset contains an indicator for insurance coverage by month. Only beneficiaries over the age of 65 by the beginning of the study period, and having full medical and pharmacy insurance coverage during the 36-month period were included . The study period is divided into a 12-month observational period, a 1 week gap period and a follow-up period of 23 months and 3 weeks . Individuals not enrolled in the Medicare Advantage program were excluded to ensure completeness of the private and public insurance data. In addition, members with fall-related diagnoses within observational or gap were excluded. For each individual, the predictive model uses claims from a 12-month observation period from the start of the study period. The outcomes (claims indicative of falls) are measured during the follow period..The study cohort comprises 119,335 beneficiaries, with 56.6% female.

The input features to the confederated machine learning model include age, gender, diagnoses as ICD 9 or ICD 10 codes, medications represented as National Drug Codes (NDC) and lab tests (encoded as LOINC codes). Lab test results were not available for this study. On average, each individual has 13.6 diagnoses, 6.9 prescriptions, and 7.4 LOINC codes during the 12 month observational period. 10,584 (8.9% ) of beneficiaries in the cohort had at least one fall in the 21-month testing period. The number of people and falls from each state are summarized in Supplementary Table 1.

### 2.2 STUDY OUTCOME

An online International Classification of Diseases, Ninth and tenth Revision (ICD-9 and ICD-10) lookup resource provided by the Centers for Medicare and Medicaid was used to obtain all codes directly associated with an accidental fall, for example, fall from a sidewalk curb (E880.1). For each member, we marked with a binary outcome variable (0 or 1) of whether a person had any fall-related claims during the follow-up period. A total of 690,295,549 medical claim diagnoses were coded in ICD-9 and 900,713,946 were in ICD-10. A total of 84 ICD-9 (Homer et al. 2017) and 330 ICD-10 (Hu Baker 2012) codes were used to identify falls.

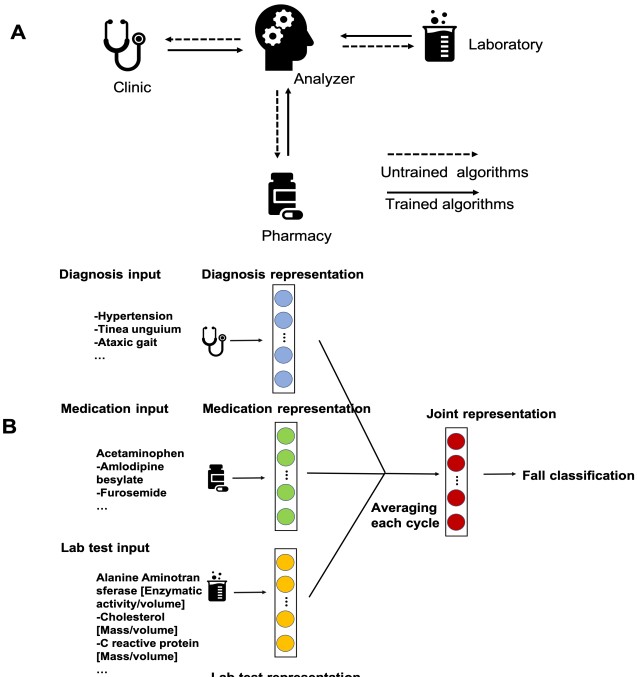

Figure 1: Confederated machine learning trains model across multiple dimensions of data separation. (A) Confederated machine learning utilizes data from clinics, pharmacies and labs simultaneously without moving the data. (B) Confederated representation joining learns joint representation from different data types in separated silos.

## 2.3 CONFEDERATED MACHINE LEARNING

For a specific state $s \in 1, 2, ...S$ with $S = 34$ in this study , Each individual $i$ has a, diagnosis vector $X_{si}^{diag}$ , a medication claim vector $X_{si}^{med}$ from pharmacy, lab test vector $X_{si}^{lab}$ from clinical lab and a binary label $Y_{si}$ , where in each state $i \in 1, 2, ...., n^s$ with $n^s$ being the number of beneficiaries in state $s$ .

At a high level, confederated learning can be intuitively understood as follows: a universal machine learning model that has both representation learning and classification components is designed for all data sites. The representation learning component takes as input a variety of input features drawn from different types of data, which cannot be presumed to all be present at the same time (Figure 1 and 2). We took the following steps:

Step 1..Claims for diagnoses, medications and lab tests during the observation period are the input features. The output of the classifier is a binary variable indicating whether the beneficiary had a fall during the follow up period. We simulate horizontal and vertical separation by separating the data for beneficiaries by U.S. state of residence. We simulate vertical separation by assuming that beneficiaries' diagnoses are only available in simulated clinics, medication claims data are only kept in simulated pharmacies and lab data only in simulated labs. Data are presumed to not be shared among different organizations nor across state lines. In total, we simulated data distributed across 102 distinct nodes including 34 clinic nodes,34 pharmacy nodes and 34 lab nodes.

Step 2. For the distributed model training, each site was delivered an array of binary target labels (fall or no fall during the 2 year follow-up period), linkable to the individual beneficiary.

Step 3. A neural network model $f$ is designed.Two different branches of input neurons (layer $\zeta_1$ ) and their directly linked connection to the next fully connected hidden layer (layer $\zeta_2$ ) were used as learning subcomponents for each input data type (Figure 1 and 2, and see next section for details): medications or lab tests. After the two branches merges, the second hidden serve as a high level representation learning and joining layer (layer $\zeta_3$ ) to integrate representation learned from

different data types. The third layers is a classifier layer (layer $\zeta_4$). it is worth pointing out that the representation and classifying power may be not completely separated by each layer in the neural network models. The parameter $\Theta$ of f is randomly initialized.

Step 4. The confederated learning model is trained as shown in Figure 1 and 2. Models with identical structures and parameters are sent to all the 102 nodes together with the binary label of falls in the follow-up period. When the machine learning model is trained on a specific data type, such as medication claims, only the representation learning subcomponents for that data type are active and subcomponents for other data types are frozen. This is implemented by sending a data availability indicator to each data source to indicate whether a certain data type $d$ is available at each site. For example, $\omega^{Pharmacy} = [1, 0]$ can be used to indicate that medication data is available at a pharmacy but lab data and diagnoses are not, where $\omega_{med}^{Pharmacy} = 1$, $\omega_{diag}^{Pharmacy} = 0$ and $\omega_{lab}^{Pharmacy} = 0$. $\omega$ is used as input into layer $\zeta_3$.

Step 5. We aggregated trained parameters from all sites by averaging the parameters to produce the updated model. By doing this, the data type specific components($\zeta_1$ and $\zeta_2$, joint representation component ($\zeta_3$) and classification components ($\zeta_4$) were learned simultaneously from different data types and individuals from all 102 sites. Information sharing among vertical separated data is achieved by joining the the representation at $\zeta_3$.

Step 6. After model aggregation, the updated model was sent out to all 102 sites again to repeat the global training cycle.

## 3 CONFEDERATED LEARNING METHOD DETAILS

The key idea behind our approach is to jointly train a representation using different data types from different sources in a distributed manner with our moving or aggregated the original data. In this section, we formally describe data organization, confederated training and inference.

The goal of confederated training of the classification model is to minimize the binary cross entropy, a metric for binary classification error, without moving any data out from from the beneficiaries' state of residency nor their data silos (pharmacies or labs). The objective function to minimize for classification model $f(X^{diag}, X^{med}, X^{lab}, \Theta)$ is:

$$L(X^{diag}, X^{med}, X^{lab}, \Theta) = \sum_{s=1}^{S} \sum_{i=1}^{n_s} -(Y_{si}log(f(X_{si}^{diag}, X_{si}^{med}, X_s i^{lab}, \Theta))$$
$$+(1 - Y_{si})log(1 - f(X_{si}^{diag}, X_{si}^{med}, X_{si}^{lab}, \Theta)))$$

Where $\Theta$ is the parameter of model $f$.

As data were not allowed to be moved out from their silos, it is not possible to train $f$ by minimizing $L(X^{d}iag, X^{m}ed, X^{l}ab, \Theta)$ in a centralized manner. Therefore, we randomly initialized a the parameter $\Theta$ as $\Theta_0$ and sent model $f$ and parameter $\Theta_0$ to pharmacies or clinical labs in each state $s \in S$.

In the clinic or hospital system of state s, we set the value of the pharmacy inputs to $0^{med}$, a zero vector, and value of the diagnoses inputs to $0^{diag}$.

The loss function is then calculated as:

$$L(X_s^{diag}, 0^{med}, 0^{lab}, \Theta_{st}^{diag}) = \sum_{i=1}^{n_s} -(Y_{si}log(f(X_s^{diag}, 0^{med}, 0^{lab}, \Theta_{st}^{diag}))$$
$$+(1 - Y_{si})log(1 - f(X_s^{diag}, 0^{med}, 0^{lab}, \Theta_{st}^{diag})))$$

Using stochastic gradient descent to minimize the loss, parameter $\Theta_{st}^{diag}$ is obtained. $t \in 1, 2....T$ stands for number of global loops which will be explained in detail just below.

In the pharmacy system of state $s$, we set the value of the lab inputs to $0^{lab}$ and value of the diagnoses inputs to $0^{diag}$, The loss function is then calculated as:

$$L(0^{diag}, X_s^{med}, 0^{lab}, \Theta_{st}^{med}) = \sum_{i=1}^{n_s} -(Y_{si}log(f(0^{diag}, X_s^{med}, 0^{lab}, \Theta_{st}^{med}))$$
$$+(1 - Y_{si})log(1 - f(0^{diag}, X_s^{med}, 0^{lab}, \Theta_{st}^{med})))$$

As above, $\Theta st^{med}$ is obtained using SGD.

In the lab system of state s, we set the value of the pharmacy inputs to $0^{med}$,and value of the diagnoses inputs to $0^{diag}$.

The loss of function in the lab system is calculated as:

$$L(0^{diag}, 0^{med}, X_s^{lab}, \Theta_{st}^{lab}) = \sum_{i=1}^{n_s} -(Y_{si}log(f(0^{diag}, 0^{med}, X_s^{lab}, \Theta_{st}^{lab}))$$
$$+(1 - Y_{si})log(1 - f(0^{diag}, 0^{med}, f(X_s^{lab}, \Theta_{st}^{lab})))$$

As above, $\Theta st^{lab}$ is obtained using SGD.

After $\Theta_{st}^{diag}$,$\Theta_{st}^{med}$ and $\Theta_{st}^{lab}$ were trained locally in each single state, they were sent back to the analyzer for aggregation by weighted averaging: $\Theta_t = \frac{1}{3S} \sum_{s=1}^{S} \frac{n^s}{N}(\Theta_{st}^{diag} + \Theta_{st}^{med} + \Theta_{st}^{lab})$ where $N$ is the total number of beneficiaries included in the study from all states. $\Theta_t$ is then sent back to clinics, pharmacies and labs in each state to repeat the whole global cycle to obtain $\Theta_{t+1}$. It is worth pointing out that in this study zero vectors were used as placeholders for data types that were not available because the performance was best on validation set when comparing with a random [0,1] vector and an all ones vector. It is absolutely possible that more sophisticated placeholders, such as vector estimated from sample distribution, would lead to better performance. However, this is not the focus of exploration in this study.

Artificial neural networks are used as the primary machine learning model for predicting fallst. The model is constructed in Keras 2.0 environment using tensorflow 1.7 as backend. Adam is used as the optimizer method with default setting (Diederik P. Kingma 2014). The model consists of three branches, one for each data type (Figure 1 and 2). Each branch consists of an input and a fully connected hidden layer with 256 neurons. The two branches merged after the hidden layer and are connected to another fully connect hidden layer with 128 neurons before the output layer with 1 neuron. The activation function used for hidden layers is "ReLu" and for output layer is "Sigmoid". In the confederated learning, 10 epochs of training are conducted during each local training cycle and global cycles are stopped when performance did not improve for 3 cycles. All the model architecture and hyperparameters were determined by grid searching based on performance on validation set. When training on clinical lab data, the model parameters corresponding to branch of medications was frozen and vice versa.

20% of randomly chosen beneficiaries were reserved as test data, and not included in the training set, 20% were chosen as validation set to adjust hyperparameters and 60% were used as training set. When conducting federated or confederated learning, data of 20% of beneficiaries from each node were used as validation set. After hyperparameter tuning, both training set and validation set were used to train the model to test performance. When testing performance of each model, the test set has centralized data with both medication claims and lab tests. Outputs from ensemble learning models were averaged to give the combined prediction.

Performance evaluation included area under the receiver operating characteristic curve (AUCROC) and area under the precision recall curve (AUCPR), AUCPR was used because the data are imbalanced– there are many more people without falls than those with falls. Instead of following the common practice of choosing a threshold that sets the false positive rates to be equal to the false negative rate (equal error rate), we chose the threshold which is 5% quantile of the predicted score of true fall. We sought to favor a screening strategy and are willing to tolerate some false positives. Using this threshold, the positive predictive value (PPV) and negative predictive value (NPV) were calculated (Table 1) and used as performance metrics in addition to AUCROC and AUCPR. Interpretation of

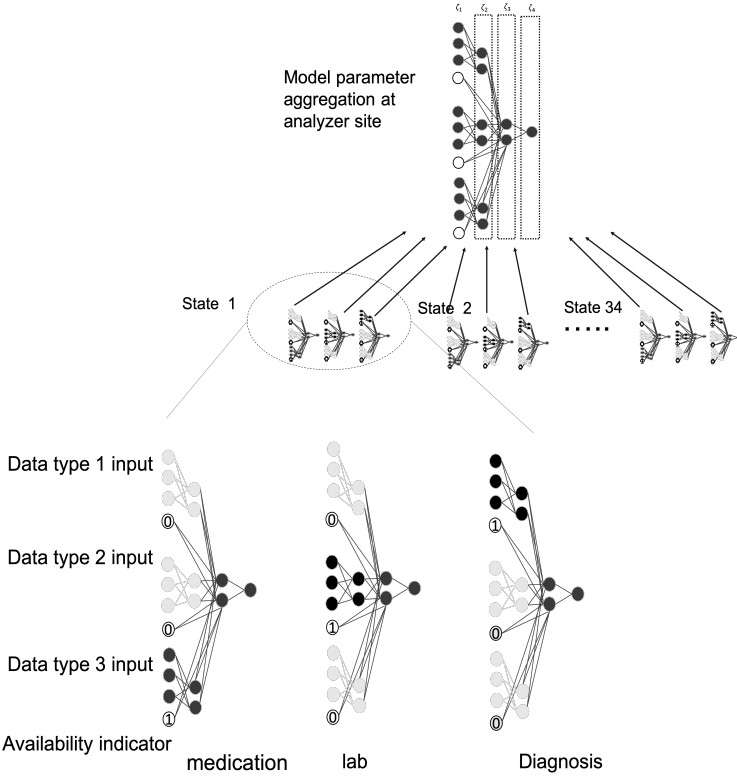

Figure 2: The confederated machine learning train a model where an individual's data are vertically separated in nodes.

the machine learning was conducted using DeepExplain package in Python2.7 using gradient input method.

## 4 EXPERIMENTAL RESULTS

We conducted experiments to compare the performance of predictive models trained in the following settings (table 1): (1) centralized learning, where data were not separated at all; (2) federated learning,where data were only horizontally separated (3) model trained on each single data type, where data were only vertically separated; (4) confederated learning, where data were both horizontally and vertically separated.

The performance of all models in this study are defined as how accurately the model can predict the fall in a testing set of randomly selected individuals from the whole study cohort that were not included in any model training, using features corresponding to the model inputs.

### 4.1 CENTRALIZED LEARNING

When we conducted the model training on aggregated data, where all types of data were centralized, the model achieved AUCROC of 0.70, AUCPR 0.21, PPV of 0.29 and NPV of 0.90. The centralized learning performance using each single data type was calculated (Table 1).

### 4.2 FEDERATED LEARNING

We note that when training on data that is horizontally but not vertically separated, confederated learning is mathematically identical to traditional federated learning. When we conducted federated learning on horizontally separated data, in which data were distributed in 34 states of residency but

Table 1: Experimental results on machine learning models for fall prediction under "separation of data types" and "separation of individuals"

| | AUCROC | AUCPR | PPV | NPV |
|---|---|---|---|---|
| **Data with no separation (centralized)** | | | | |
| Learning on aggregated data | 0.7 | 0.21 | 0.29 | 0.9 |
| **Data horizontally separated** | | | | |
| Federated learning | 0.68 | 0.2 | 0.31 | 0.9 |
| **Data vertically separated** | | | | |
| Diagnosis data only | 0.67 | 0.19 | 0.28 | 0.9 |
| Medication claim data only | 0.64 | 0.17 | 0.22 | 0.9 |
| Lab test record only | 0.59 | 0.14 | 0.15 | 0.9 |
| Ensemble learning | 0.64 | 0.17 | 0.22 | 0.9 |
| Confederated Learning | 0.68 | 0.2 | 0.29 | 0.9 |
| **Data horizontally and vertically separated** | | | | |
| Diagnosis data only | 0.67 | 0.19 | 0.28 | 0.9 |
| Medication claim data only | 0.63 | 0.17 | 0.24 | 0.9 |
| Lab test record only | 0.6 | 0.15 | 0.18 | 0.9 |
| Ensemble learning | 0.63 | 0.17 | 0.22 | 0.9 |
| *Confederated Learning* | 0.68 | 0.21 | 0.31 | 0.9 |

not vertically separated, the model achieved an AUCROC of 0.68, AUCPR 0.20, PPV of 0.31 and NPV of 0.90.

### 4.3 CONFEDERATED LEARNING

Using Confederated Representation Joining on data that is vertically but not horizontally separated (single degree of separation), the algorithm achieved an AUCROC of 0.68, AUCPR of 0.20, PPV of 0.29 and NPV of 0.90 on predicting fall in follow-up period.

Next, we conducted experiments to show that confederated learning is able to train a distributed model using distributed data with two degrees of separation, both horizontally and vertically. The confederated learning algorithm achieved an AUCROC of 0.68, AUCPR of 0.21, PPV of 0.31 and NPV of 0.90 on predicting fall, which is comparable to centralized learning with all data aggregated and to federated learning where data were only horizontally separated. Performances of confederated representation joining in both vertical separation and vertical plus horizontal separation were better with ensemble learning where output from model trained on different data types were averaged.

In order to understand behaviours of trained predictive models, the importance of each feature in the predictive models was calculated. The ten most important variables for the machine learning model trained in centralized learning are shown in supplementary table 2 The ten most important variables for the machine learning model trained in confederated learning are shown in supplementary table 3. Hypertension, edema, and movement related medical conditions are found in both lists, which suggests the two models work in similar manners.

### 5 CONCLUSION AND DISCUSSION

Currently, the clinical screening process generally involves asking patients 65 years old and above questions about their previous falls and walking balance. People who give positive answers to the question can be further assessed for their balance and gait (Panel on Prevention of Falls in Older Persons, American Geriatrics Society and British Geriatrics Society 2011). Though the guidelines-based clinical assessment reduces falls, it is costly and time consuming to conduct large scale screening. A machine learning approach can stratify individuals by risk of fall, using their electronic health record data.

We demonstrate that health data distributed across silos can be used to train machine learning models without moving or aggregating data, even when data types vary across more than one degree of separation. Compared with other methods for model training on horizontally and vertically separate

data, this confederated learning algorithm does not require sophisticated computational infrastructure , such homomorphic encryption, nor frequent gradient exchange.

We anticipate that this confederated approach can be extended to more degrees of separation. Other type of separation, such as separation by temporality , separation by insurance plan, separation by healthcare provider can all be potentially be explored using confederated learning strategy. One such example of additional degree of separation is a patient's diagnosis might be distributed with different healthcare providers or his/her medication information is with more than one pharmacy

---

**Algorithm 1** Confederated Representation joining

---

**Input:** Medications claims data ($X^{med}$), lab tests records ($X^{lab}$) distributed in $S$ states and binary labels of fall ($Y$)

Parameter and hyperparameters of the neural networks $\Theta$ **Output:** Whether an elderly will fall in follow-up period

Initialize neural network model $f$ with parameter $\Theta_0$

**for** $t \in 1$ **to** $T$ *in parallel* **do**

  // $T$ is total number of global cycles

  **for** *State* $s \in 1$ **to** $S$ *in parallel* **do**

    // Conducted in parallel across all $S = 34$ states

    In clinic node:

    $\Theta_{st}^{diag} \leftarrow \Theta_{t-1}$ // identical model parameters sent to each site

    Lab test branch of layers $\zeta_1$ and $\zeta_2$ are frozen

    fit $f$

    Obtain parameters of $f$ as $\Theta_{st}^{diag}$

    In pharmacy node:

    $\Theta_{st}^{med} \leftarrow \Theta_{t-1}$ // identical model parameters sent to each site

    Lab test branch of layers $\zeta_1$ and $\zeta_2$ are frozen

    fit $f$

    Obtain parameters of $f$ as $\Theta_{st}^{med}$

    In lab node:

    $\Theta_{st}^{lab} \leftarrow \Theta_{t-1}$ // identical model parameters sent to each site

    Medication branch of layers $\zeta_1$ and $\zeta_2$ are frozen

    Fit $f$

    Obtain parameters of $f$ as $\Theta_{st}^{lab}$

  **end**

  Update model parameter by $\Theta_t = \frac{1}{3S} \sum_{s=1}^{S} \frac{n_s}{N} (\Theta_{st}^{diag} + \Theta_{st}^{med} + \Theta_{st}^{lab})$

  // Update model parameters using weighted average from model parameters of each site

  Set parameters of $f$ as $\Theta_t$

  $n_s$ is the number of patients at state $s$ and N is the total population size across all states

**end**

---

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

## A    APPENDIX

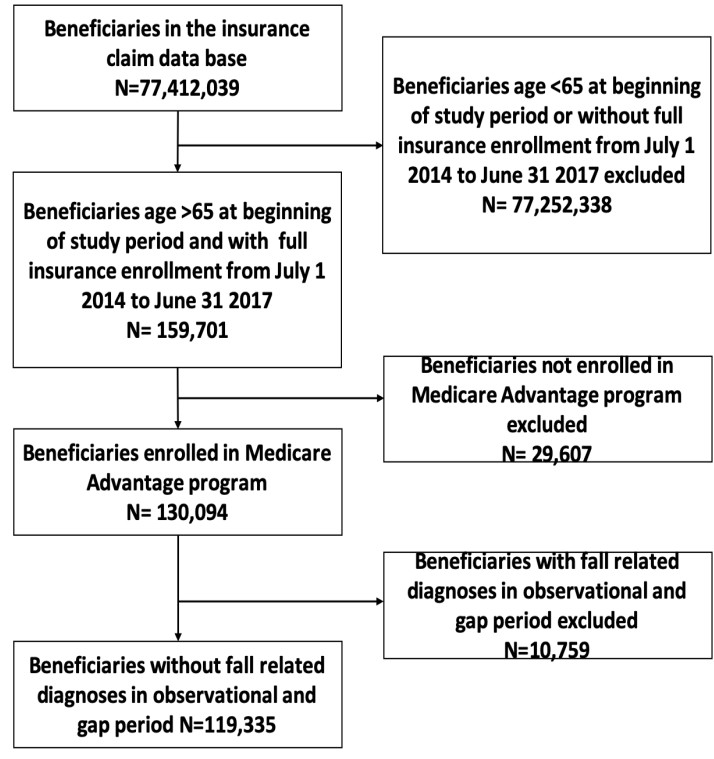

Figure 3: Study cohort selection process from the health insurance claim database.

Table 2: Top 10 variables in model trained on centralized data

| varaible names | Mean Weights | variable type |
|---|---|---|
| 401.1 | 0.0032 | diag |
| 729.5 | 0.0026 | diag |
| 110.1 | 0.002 | diag |
| V76.12 | 0.002 | diag |
| 401.9 | 0.0017 | diag |
| V70.0 | 0.0017 | diag |
| 781.2 | 0.0015 | diag |
| 530.81 | 0.0015 | diag |
| 782.3 | 0.0014 | diag |
| atorvastatin | 0.0014 | med |

Table 3: Top 10 variables in model trained in a confederated manner

| varaible_names | Mean_Weights | variable_type |
|---|---|---|
| 401.9 | 0.0058 | diag |
| V70.0 | 0.0046 | diag |
| 729.5 | 0.0043 | diag |
| V76.12 | 0.0036 | diag |
| furosemide | 0.0034 | med |
| acetaminophen | 0.0031 | med |
| 110.1 | 0.0029 | diag |
| 366.16 | 0.0028 | diag |
| 782.3 | 0.0028 | diag |
| 285.9 | 0.0028 | diag |

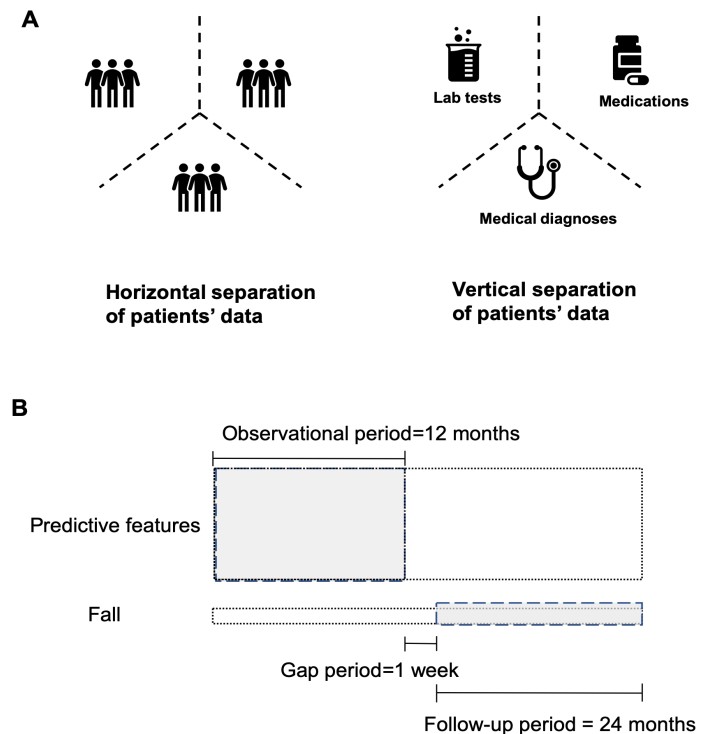

Figure 4: (A)Two degrees of separation. Horizontal separation refers to fragmentation of an individual's data across silos, for example across hospitals and clinics. Vertical separation refers to differences in the domain, semantics and structure of the data, for example, data from pharmacies, clinics and labs, each in their own nodes. (B) Study period. Patient's data are divided into three periods. Observational period is 12 months, gap period is 1 week and follow-up period is 21 months. Diagnosis, medication and lab data from observational period are used as predictive features for fall in follow-up period. The 1-week gap period is introduced to avoid complication of encounters happened directly before fall.

