# OpenReview forum: "CONFEDERATED MACHINE LEARNING ON HORIZONTALLY AND VERTICALLY SEPARATED MEDICAL DATA FOR LARGE-SCALE HEALTH SYSTEM INTELLIGENCE"
_ICLR.cc/2020/Conference — Reject_

### Official Review · AnonReviewer3 · 2019-10-22
**Official Blind Review #3**

**Rating:** 3

**Review:**

This paper considers the problem of learning from medical data that is separated both horiontally (across different practices and centers) and vertically (by data type). The contribution is a "confederated" machine learning method that learns across these divides. The particular application considered here as means of illustration is that of fall prediction in the elderly. Specifically the authors investigate an ML approach to risk-stratifying elderly patients with respect their likelihood of falling in the next two years.

The basic challenge addressed here is learning in the setting in which different data elements are available only at specific sites, and it is assumed that they do not share data. In addition, it is assumed there are multiple distinct sites that have data corresponding to the respective elementes. However, it is assumed that labels (target vectors) are shared across all sites. This setup is simulated using available data. A simple distributed training scheme is outlined.

This is a potentially important problem worthy of study. I some major concerns with the present work, however. First, I do not think that ICLR is really the best venue for this work. The machine learning component of this is quite straightforward; basically SGD is performed iteratively on parameters associated with the data types "owned" by the respective (simulated) sites. Updates are then averaged over these parameter subsets. This is perfectly reasonable, but not terribly novel. The presentation of this is also much longer than it needs to be for the ICLR audience. I think this paper, in its current form, would be better suited for an audience more interested in clinical applications specifically (and I say this as someone quite appreciative of work on applied ML; it's just that the audience here will be more interested in methodological innovations.)

With respect to clinical utility: Do we really need ML to tell us about risk of falls? I mean, if we were to ask the MD who had seen these patients to perform a simple stratification (perhaps on an ordinal scale), would they not likely be able to do so reasonably well? The authors mention something like this, discussing the 'clinical screening process' which involves asking about prior falls. This seems like a really strong baseline. The authors argue that this is time-consuming,

In any case, is AUCPR an appropriate or useful metric here? In practice one would need to pick a threshold on which to act; perhaps a simulation that investigated doing so would provide a more meaningful evaluation. Although again a strong baseline here would probably be to ask physicians to risk stratify patients for interventions direclty (I appreciate that this would be a non-trivial experiment to run, but still).

I also have a question regarding the simulation. I *think* the authors have randomly assigned patients to the respective simulated sites; is that right? This seems problematic because in practice patients would not be IID distributed in this way; sites would have their own patient populations which would affect the losses. This should be somehow taken into account in the simulation.


Other comments
---
- I think I am missing something in the notation here. $Y_{si}$ is a 'binary label' but seems to vary across 'states' for an individual, is that right? Shouldn't this be constant for an individual? The paper states below that "The output of the classifier is a binary variable indicating whether the beneficiary had a fall during the follow up period."

- Labels were derived from ICD codes; was there any effort to spot check these? I am always a bit concerned about deriving labels from ICD and trusting them.

- As far as I understand from 2.1, the authors have not included features extracted from notes in the patient history; is that right? Why not?

Smaller issues
---
- I would strongly suggest numbering your equations. Also, suggest using \text while in mathmode for superscripts like `diag'.

- "Step 1.." --> "Step 1." (p3)"

- "The parameter Θof f is randomly initialized" --> missing space before "of"

- On page 4: L(Xdiag, Xmed, Xlab, Θ) is written incorrectly.

- page 4: "Tstands"  missing space.

- page 5: " fallst." --> "falls."

- Appendix Tables 2 and 3 both contain the typo "varaible" (should be "variable")

- In Appendex Table 2, I suggest reporting results with a consistent amount of precision, e.g., 0.002 --> 0.0020 here.

**Experience Assessment:**

I have published one or two papers in this area.

**Review Assessment: Checking Correctness Of Derivations And Theory:**

I assessed the sensibility of the derivations and theory.

**Review Assessment: Checking Correctness Of Experiments:**

I assessed the sensibility of the experiments.

**Review Assessment: Thoroughness In Paper Reading:**

I read the paper at least twice and used my best judgement in assessing the paper.

---

### Official Review · AnonReviewer2 · 2019-10-23
**Official Blind Review #2**

**Rating:** 1

**Review:**

The authors propose a learning strategy to fit predictive models on data separated across nodes, and for which different set of features are available within each node.
This concept is developed by introducing the concept of two degree separation across horizontal (nodes) and vertical (feature) axis. The proposed approach consists in an iterative scheme where i)  models at independentently trained at each site, and ii) models' parameters are subsequently averaged and redistributed for the next optimisation round.

The problem tackled in this work is interesting, with an important application on medical records from > 100,000 individuals followed  over time. Unfortunately the paper is not clear in several aspects, and presents methodological issues. Here my main comments on this work:

- The authors should definitely refer to the concept of meta-learning [1], which addresses modelling problems very close to the one presented in this work: training a meta-model by aggregating information from different learning tasks.  The paper should definitely compare the proposed methodology with respect to this paradigm.

- The fact that the parameters can be averaged across nodes implies that they must be of same dimension. This is counterintuitive, as the dimension of the data represented at each site may significantly differ depending on the kind of considered feature. This aspect points to some methodological inconsistency.

- There is no comparison with any other federated method, neither with any classification method besides a NN, at least with the aggregated data. Also it could have been possible to reduce the number of input features using simple dimensionality reduction previous to the NN, such as PCA.

- Vertical separation importance: At the end it looks like diagnosis is the main driver for the classification, showing results that are comparable to the ones obtained with the aggregated data. It is therefore not clear whether the proposed application allows to clearly illustrate the benefit of using this method with regard to vertical separation.

- All in all, the paper appears in a draft form, and the text is often inconsistent. For example, there is often inconsistency in the number of branches, or types of data considered, figures are not self-explanatory and present notation and symbols not defined anywhere. The bibliography is given in a non-standard format.

[1] Model-Agnostic Meta-Learning for Fast Adaptation of Deep Networks. Finn, C., Abbeel, P., & Levine, S.  Proceedings of the 34th International Conference on Machine Learning-Volume 70 (pp. 1126-1135).

**Experience Assessment:**

I have published one or two papers in this area.

**Review Assessment: Checking Correctness Of Derivations And Theory:**

I carefully checked the derivations and theory.

**Review Assessment: Checking Correctness Of Experiments:**

I carefully checked the experiments.

**Review Assessment: Thoroughness In Paper Reading:**

I read the paper thoroughly.

---

### Official Review · AnonReviewer1 · 2019-10-24
**Official Blind Review #1**

**Rating:** 1

**Review:**

This paper describes a structure to approach the federated machine learning problem for hospitals. The approach does not seem very novel and it is hard to see what the representation learning challenges are. There is no open benchmark that the community can work on.

I suggest that the paper focus on the method and not the private dataset used. If you cannot release a public dataset then maybe a synthetic dataset that presents known challenges you observe in private data. This can be used as a benchmark for the community to improve these methods.

Typos:

 "Step 1..Claims"

Some of the citations seem to be listing every author of the paper which is very hard to read the paper.


**Experience Assessment:**

I have published in this field for several years.

**Review Assessment: Checking Correctness Of Derivations And Theory:**

I assessed the sensibility of the derivations and theory.

**Review Assessment: Checking Correctness Of Experiments:**

I assessed the sensibility of the experiments.

**Review Assessment: Thoroughness In Paper Reading:**

I read the paper at least twice and used my best judgement in assessing the paper.

---

### Decision · Program_Chairs · 2019-12-19

**Decision:**

Reject

**Comment:**

This manuscript proposes a strategy for fitting predictive models on data separated across nodes, with respect to both samples and features.

The reviewers and AC agree that the problem studied is timely and interesting, and were impressed by the size and scope of the evaluation dataset (particularly for a medical application). However, reviewers were unconvinced about the novelty and clarity of the conceptual and empirical results. On the conceptual end, the AC also suggests that the authors look into closely related work on split learning (https://splitlearning.github.io/) which has also been applied to medical data settings.